# Trends in Caesarean Section Rate According to Robson Group Classification among Pregnant Women with SARS-CoV-2 Infection: A Single-Center Large Cohort Study in Italy

**DOI:** 10.3390/jcm11216503

**Published:** 2022-11-02

**Authors:** Angelo Sirico, Luigi Carbone, Luisa Avino, Cira Buonfantino, Maria Chiara De Angelis, Marco Di Cresce, Annamaria Fabozzi, Francesco Paolo Improda, Antonietta Legnante, Carla Riccardi, Romina Santoro, Roberta Vallone, Brunella Zizolfi, Antonio Riccardo Buonomo, Ivan Gentile, Serena Salomè, Francesco Raimondi, Giuseppe Bifulco, Maurizio Guida

**Affiliations:** 1Department of Neuroscience, Reproductive Sciences and Dentistry, University of Naples Federico II, 80131 Naples, Italy; 2Department of Clinical Medicine and Surgery, University of Naples Federico II, 80131 Naples, Italy; 3Division of Neonatology, Section of Pediatrics, Department of Translational Medical Sciences, University of Naples Federico II, 80131 Naples, Italy

**Keywords:** COVID-19, pregnancy, Robson group classification, caesarean section, high-risk pregnancy, induction of labor, delivery

## Abstract

Background: Since there is no available data on temporal trends of caesarean section (CS) rates in pregnant women with COVID-19 through the pandemic, we aimed to analyze the trends in caesarean section rate in a large cohort of pregnant women with COVID-19, according to the Robson Ten Group Classification System of deliveries. Methods: We prospectively enrolled pregnant women with a diagnosis of COVID-19 who delivered in our center between March 2020 and November 2021. Deliveries were classified, according to the Robson group classification, and according to three time periods: (1) deliveries from March 2020 to December 2020; (2) deliveries from January 2021 to April 2021; (3) deliveries from May 2021 to November 2021. We compared pregnancy characteristics and incidence of caesarean section, according to the Robson category in the total population, and according to the three time periods. Results: We included 457 patients matching the inclusion criteria in our analysis. We found that overall CS rate significantly decreased over time from period 1 to period 3 (152/222, 68.5% vs. 81/134, 60.4% vs. 58/101, 57.4%, χ^2^ = 4.261, *p* = 0.039). CS rate significantly decreased over time in Robson category 1 (48/80, 60% vs. 27/47,57.4% vs. 8/24, 33.3%, χ^2^ = 4.097, *p* = 0.043) and Robson category 3 (13/42, 31% vs. 6/33, 18.2% vs. 2/22, 9.1%, χ^2^ = 4.335, *p* = 0.037). We also found that the incidence of induction of labor significantly increased over time (8/222, 3.6% vs. 12/134, 9% vs. 11/101, 10.9%, χ^2^ = 7.245, *p* = 0.027). Conclusion: Our data provide an overview of the temporal changes in the management and obstetric outcome of COVID-19 pregnant women through the pandemic, confirming that standards of obstetrical assistance for pregnancies complicated by SARS-CoV-2 infection improved over time.

## 1. Introduction

The global pandemic caused by the coronavirus SARS-CoV-2 infection has posed strong clinical challenges for the management of patients, especially for pregnant women. Coronavirus disease-19 (COVID-19), first documented in Wuhan, China at the end of 2019 [1], has rapidly spread across the globe, infecting hundreds of millions of individuals [2].

Soon after the pandemic outbreak, cases of infected pregnant women were reported, highlighting concerns about the increased risks of maternal morbidity and neonatal adverse outcomes in these pregnancies [3,4]. Subsequently, large multi-center studies have confirmed that pregnant woman infected with SARS-CoV-2 might experience higher respiratory morbidity and mortality, while the risk of vertical transmission seems to be extremely low [5,6,7,8]. Pregnant women infected with SARS-CoV-2 have a higher risk of ICU admission, respiratory support by mechanical ventilation, maternal mortality, miscarriage, preterm birth, and delivery by caesarean section, compared to the general population and non-infected pregnant women [9,10,11].

The current situation of the pandemic, with the daily rise of reported cases all over the globe, changed the routine of healthcare practices, as new challenges appeared; these affected obstetric and childbirth healthcare services [12,13,14]. First recommendations suggested that suspected, probable, and confirmed cases of COVID-19 infection should be managed initially by designated tertiary hospitals with effective isolation facilities and protection equipment to assure fetal and maternal benefit along with minimal risks for healthcare providers [15,16,17]. Published recommendations highlighted that mode of delivery should not be influenced by the presence of COVID-19, unless the woman’s respiratory condition demands urgent intervention for birth [3,18]. Therefore, mode and timing of delivery should be individualized based on obstetric indications and maternal–fetal status. Despite that, although COVID-19 infection itself is not an indication for delivery by caesarean section, first published data showed that obstetrical care for women with COVID-19 in isolation settings was associated with increased rate of delivery by caesarean section [18,19].

Robson’s system classifies all deliveries into one of ten groups based on five parameters: obstetric history (parity and previous caesarean section), onset of labor (spontaneous, induced, or caesarean section before onset of labor), fetal presentation or lie (cephalic, breech, or transverse), number of neonates, and gestational age. The Ten Group Classification System (TGCS) is already being used in more than 50 countries and has been endorsed by the WHO, International Federation of Gynecology and Obstetrics, and the European Board of Obstetrics and Gynecology [20,21,22,23]. The ten Robson categories are mutually exclusive, totally inclusive, and can be applied prospectively, since each woman admitted for delivery can be classified immediately based on a few variables that are generally routinely recorded. This system helps institution-specific monitoring and auditing and offers a standardized comparison method between institutions, countries, and timepoints.

Since there is no available data on temporal trends of caesarean section rates in pregnant women with COVID-19 through the pandemic, we aimed to analyze the trends in caesarean section rate in a large cohort of pregnant women with COVID-19, according to the Robson TGCS.

## 2. Materials and Methods

We prospectively enrolled pregnant women with diagnosis of COVID-19 who delivered in our center between March 2020 and November 2021. Beginning from March 2020, University hospital Federico II of Naples, Italy, has been classified as referral center for pregnant women with COVID-19 for the regional area (5.8 million people). Infrastructural adaptations were made to provide an isolated route for infected patients, including a dedicated obstetric emergency room, a dedicated antenatal ward, labor ward, operating theatre, and a bio-containment area for suspected cases. Patients with a positive molecular nasal/oro-pharyngeal swab, which is a test that detects the genome (RNA) of the SARS-CoV-2 virus in the biological sample using the RT-PCR method, were admitted into the dedicated COVID-19 ward while women with fever, cough, dyspnea, positive rapid antigen-test, or contact with a COVID-19 positive person within 7 days were admitted to the bio-containment area until a molecular nasal/oro-pharyngeal swab test was available.

We included in our analysis pregnant women who were diagnosed with SARS-CoV-2 infection and delivered at our center. Exclusion criteria were presence of positive rapid antigen-test without confirmed infection with a molecular test, persistent asymptomatic infection after 21 days from the first molecular test, infected patients who were discharged from our unit before delivery, and patients who were subsequently diagnosed negative at molecular nasal/oro-pharyngeal swab before delivery, presence of fetal chromosomal, or congenital anomalies.

Data on SARS-CoV-2 vaccination were collected from March 2021 when the vaccines started to be available in Italy for pregnant women, and patients who did not undergo vaccination were asked to fill a questionnaire about the reason for this choice (allergies, previous medical conditions, fears for the vaccine safety).

For each patient, we collected the following data: maternal age, parity, last menstrual period, pregestational weight and BMI, gestational weight gain (GWG) at delivery, fetal growth restriction (FGR), pregestational and gestational diabetes, maternal hypertension (including chronic hypertension, gestational hypertension and preeclampsia), gestational age (GA) at delivery, vaccination against SARS-CoV-2, induction of labor, type of delivery (spontaneous delivery or caesarean section), indication for caesarean section, neonatal sex, and neonatal birthweight.

### Statistical Analysis

Based on maternal and pregnancy data, deliveries were classified, according to Robson TGCS (Table 1). Furthermore, patients were grouped according to three time periods: (1) deliveries from March 2020 to December 2020; (2) deliveries from January 2021 to April 2021; (3) deliveries from May 2021 to November 2021. 

We compared pregnancy characteristics and incidence of caesarean section, according to the Robson category in the total population. Subsequently, investigated variables were evaluated, according to the three time periods. Continuous data were expressed as mean and standard deviation when normally distributed or otherwise as median and interquartile range (IQR). Categorical variables were expressed as number (n) and percentage (%) of the group. The Shapiro–Wilk test was performed to test for normality. When normally distributed, continuous variables were compared by Student’s *t*-test, otherwise by Kruskal–Wallis test; categorical variables were compared by Mantel–Haenszel test for trend. We also compared the caesarean section rate in the cohort of infected women with the caesarean section rate in the same institution in 2019, before the pandemic. A *p* value < 0.05 was considered as significant. Statistical analysis was carried out using the Statistical Package for Social Sciences (SPSS) Statistics v. 19 (IBM Inc., Armonk, NY, USA).

## 3. Results

We included 457 consecutive patients matching the inclusion criteria in our analysis. Pregnancy and neonatal characteristics are summarized in Table 2. All included patients did not undergo vaccination against SARS-CoV-2 either because it was not yet available, or they did not feel reassured enough on vaccine safety in pregnancy. Included women delivered by spontaneous delivery in 166 cases (36.3%), while caesarean section was performed in 291 cases (63.7%); urgent caesarean section for worsening maternal respiratory distress was performed in 22 cases (4.8%). According to Robson TCGS, among 457 deliveries, there were 151 (33%) category 1, 23 (5%) category 2, 97 (21.2%) category 3, 10 (2.2%) category 4, 133 (29.1%) category 5, 8 (1.8%) category 6, 1 (0.2%) category 7, 6 (1.3%) category 8, 1 (0.2%) category 9, and 27 (5.9%) category 10. Among CS deliveries, there were 83 (54.9%) in category 1, 14 (60.8%) in category 2, 21 (21.6%) in category 3, 3 (30%) in category 4, 131 (98.5%) in category 5, 8 (100%) in category 6, 1 (100%) in category 7, 5 (83.3%) in category 8, 1 (100%) in category 9, and 24 (88.9%) in category 10 (Figure 1).

After grouping pregnancies, according to the three time periods of delivery, data analysis showed no significant difference between groups in maternal age, pregestational BMI, GWG, neonatal sex, incidence of neonatal birthweight < 2500 g and <1500 g, incidence of FGR, maternal hyperglycemia, or hypertension (Table 3). Incidence of induction of labor significantly increased over time from period 1 to period 3 (8/222, 3.6% vs. 12/134, 9% vs. 11/101, 10.9%, χ^2^ = 7.245, *p* = 0.03), as well as incidence of preterm birth (24/222, 10.9% vs. 20/134, 15.3% vs. 21/101, 20.8%, χ^2^ = 5.656, *p* = 0.02) and incidence of urgent CS performed for worsening maternal respiratory distress (8/222, 3.6% vs. 4/134, 3% vs. 10/101, 9.9%, χ^2^ = 7.392, *p* = 0.03).

We found that overall CS rate significantly decreased over time from period 1 to period 3 (152/222, 68.5% vs. 81/134, 60.4% vs. 58/101, 57.4%, χ^2^ = 4.261, *p* = 0.03). CS rate significantly decreased over time in Robson category 1 (48/80, 60% vs. 27/47,57.4% vs. 8/24, 33.3%, χ^2^ = 4.097, *p* = 0.04) and Robson category 3 (13/42, 31% vs. 6/33, 18.2% vs. 2/22, 9.1%, χ^2^ = 4.335, *p* = 0.03). We also found that CS rate significantly increased over time in Robson category 10 (7/10, 70% vs. 8/8, 100% vs. 9/9, 100%, χ^2^ = 4.291, *p* = 0.03) (Figure 2). 

In 2019, before the pandemic, our institution delivered 2443 pregnancies and 1151 underwent a caesarean section. CS rate was lower in the non-infected women before the pandemic, compared to the total cohort of COVID-19 patients (1151/2243, 47.1% vs. 291/457, 63.7%, χ^2^ = 41.584, *p* < 0.001) but also compared to COVID-19 patients in the time period 1 (151/222, 68.5%, χ^2^ = 36.143, *p* < 0.001) and time period 2 (81/134, 60.4%, χ^2^ = 8.524, *p* = 0.003), while no significant difference was found between CS rate before the pandemic and CS rate in time period 3 (58/101, 57.4%, χ^2^ = 3.731, *p* = 0.0534).

## 4. Discussion

Our data show that, in a population of pregnant women infected by SARS-CoV-2, caesarean section rate significantly decreased through the pandemic. When considering deliveries, according to Robson category, caesarean section rate also significantly decreased in Robson category 1 and category 3 and increased in category 10. 

This is the first study to investigate temporal trends in caesarean section rate and pregnancy outcomes through the pandemic in a population of women with COVID-19. Parazzini et al. reviewed the first cases of COVID-19 pregnant women in early 2020 and, among 64 included patients, CS was performed in 90.6% of cases [3]. Data from another group of 68 infected women who delivered from December 2019 to March 2020 in China also showed a CS rate of 93%, confirming the increased caesarean section rate among infected pregnant women who delivered in the first wave of SARS-CoV-2 pandemic [19]. A review from Debrabandere included 203 cases, of whom 140 delivered by CS (68.9%), while Prabhu et al. found that SARS-CoV-2 positive women were more likely to undergo caesarean delivery than women testing negative (46.7% in symptomatic COVID-19, 45.5% in asymptomatic COVID-19, and 30.9% in women without COVID-19; *p* = 0.044) [24,25]. Knight M also reported 59% of CS rate in a cohort of women with COVID-19 between March and April 2020 in the UK [26]. Gandhi et al. conducted an observational study among 53 hospital departments in India from April to August 2020, including 771 pregnant women who delivered during the study period; data analysis showed a CS rate of 59.01% [27]. On the other hand, an analysis of women delivering at New York City hospitals between March and April 2020 found CS rates no higher than average (31.3% for women with confirmed COVID-19, compared to 33.9% of those who tested negative) [28].

A multicenter cohort study from the World Association of Perinatal Medicine Working Group on COVID-19 conducted in 72 centers worldwide included 251 women with a singleton pregnancy who tested positive for SARS-CoV-2 infection and delivered from February to April 2020; data analysis showed an overall CS rate of 54.2%, which was higher among symptomatic women, compared to asymptomatic cases (56.5% vs. 48.6%) [10]. 

Furthermore, a population-based cohort study in England from May 2020 to January 2021, including 342,080 singleton pregnancies of which 3527 confirmed SARS-CoV-2-infected women demonstrated that pregnant women with COVID-19 were more likely to undergo emergency caesarean delivery (27.6% vs. 18.5%), compared to non-infected pregnant women [29]. 

Data from the literature confirmed an increased CS rate among infected pregnant women, which is comparable to our results. However, none of the studies published so far evaluated the incidence of CS, according to the Robson group classification, probably because these studies were conducted on small cohorts or included heterogeneous data from different hospital centers. This is the first study to analyze CS rate among COVID-19 pregnant women, according to the Robson group classification in a large single-center study, thus, providing a more accurate evaluation of the obstetrical interventions, according to maternal anamnestic data. 

Previous authors did not explain the causes of the increased CS rate among women with COVID-19. Personnel healthcare assistance to infected women in labor at the beginning of the pandemic was challenging; the approach to a never-before experienced situation, the concerns about the possible vertical transmission, the recommendations to expedite delivery, the stressful condition to assist women wearing full PPE in a restricted bio-contained area during the labor, and the risk of performing emergency procedures in a contaminated area were probably at the basis of the decision to shift for CS in case of any uncertainty. 

Our hypothesis is that assisting pregnant women with SARS-CoV-2 infection over time may have increased personnel experience and confidence to work in a bio-contained restricted area and to provide an obstetric assistance comparable to a standard labor ward. This hypothesis may explain the significant decrease in CS rate in our cohort through the pandemic, especially for nulliparous and multiparous women without the induction of labor (Robson group 1 and 3), for whom the CS rate declined, respectively from 60% to 33.3% and from 31% to 9.1%. In the same way, our data showed that induction of labor significantly increased over time from 3.6% to 10.9%, thus, confirming the increased confidence of healthcare personnel in managing infected pregnant women. Data from our cohort also highlighted an increased CS rate in Robson group 10 in the period from May to November 2021, coincident with an increase in preterm birth and urgent CS performed for worsening maternal respiratory distress. These results are consistent with the increased morbidity observed in pregnancies with COVID-19 during the recent surge associated with the Delta variant, particularly in a pregnant population where vaccine acceptance is low [30,31,32,33,34]. 

Data from this study confirm that mode of delivery should not be influenced by the presence of COVID-19. With time, the delivery management of infected pregnant women without respiratory distress results in CS rates similar to that reported for non-infected women.

This is the first study to investigate trends in caesarean section rate through the pandemic in a large population of women with SARS-CoV-2 infection from a single center. Furthermore, no previous data are available on the evaluation of caesarean section rate, according to Robson group classification. 

Limitations of the study include lack of assessment of potential confounders and anamnestic assessment and the lack of a control group of women without COVID-19. Even if a comparison with a control group was not provided, it is noteworthy that our institution CS rate in 2019 was lower than CS rate among COVID-19 patients, except for the last time period, confirming that, ultimately, the obstetric management for COVID-19 patients became similar to non-infected patients. Furthermore, our study includes only a small group of pregnant women who underwent the induction of labor. Therefore, more data are needed on the obstetric outcome of the induction on labor of infected women.

## 5. Conclusions

Our data provide an overview of the temporal changes in the management and obstetric outcome of COVID-19 pregnant women through the pandemic, confirming that obstetric assistance for pregnancies with SARS-CoV-2 infection improved over time.

## Figures and Tables

**Figure 1 jcm-11-06503-f001:**
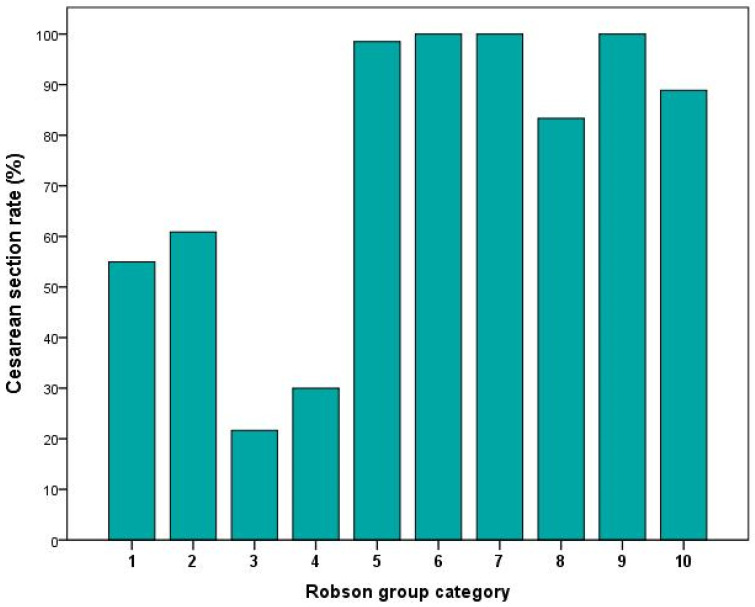
Caesarean section rate of included COVID-19 pregnancies, according to the Robson group category.

**Figure 2 jcm-11-06503-f002:**
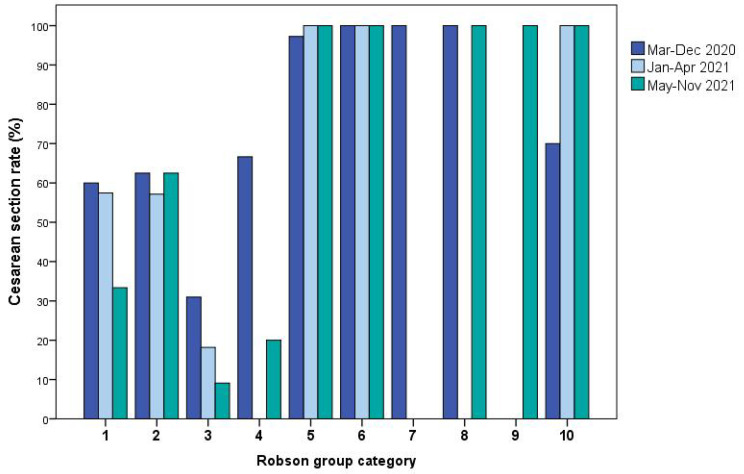
Caesarean section rate of included COVID-19 pregnancies according to the Robson group category and the time period. Missing columns for groups 7 and 9 are referred to absence of cases in the period.

**Table 1 jcm-11-06503-t001:** The Robson Ten Group Classification System of deliveries.

Robson Group	Description
Group 1	Nulliparous, single cephalic, ≥37 weeks, in spontaneous labor.
Group 2	Nulliparous, single cephalic, ≥37 weeks, induced or CS before labor.
Group 3	Multiparous (excluding previous caesarean section), singleton, cephalic, ≥37 weeks’ gestation, in spontaneous labor.
Group 4	Multiparous without a previous uterine scar, with singleton, cephalic pregnancy, ≥37 weeks’ gestation, induced or caesarean section before labor.
Group 5	Previous caesarean section, singleton, cephalic, ≥37 weeks’ gestation.
Group 6	All nulliparous with a single breech.
Group 7	All multiparous with a single breech (including previous caesarean section).
Group 8	All multiple pregnancies (including previous caesarean section).
Group 9	All women with a single pregnancy in transverse or oblique lie (including those with previous caesarean section).
Group 10	All singleton, cephalic, <37 weeks’ gestation pregnancies (including previous caesarean section).

**Table 2 jcm-11-06503-t002:** Pregnancy characteristics of included patients. Data presented as n (%) or median [interquartile range].

	N = 457	*p* *
Maternal age	30 [8]	0.004
Maternal pregestational BMI	30.1 [6.7]	<0.001
Maternal GWG	11.25 [6]	<0.001
FGR	24 (5.2%)	
Maternal hyperglycemia	36 (7.9%)	
Maternal hypertension	30 (6.6%)	
pPROM	9 (2%)	
Induction of labor	31 (6.8%)	
GA at delivery	39 [2]	<0.001
Preterm birth	65 (14.3%)	
Neonatal weight < 2500	48 (10.5%)	
Neonatal weight < 1500	14 (3.1%)	
CS delivery	291 (63.7%)	
Urgent CS for respiratory distress	22 (4.8%)	
Robson category 1	151 (33%)	
Robson category 2	23 (5%)	
Robson category 3	97 (21.2%)	
Robson category 4	10 (2.2%)	
Robson category 5	133 (29.1)	
Robson category 6	8 (1.8%)	
Robson category 7	1 (0.2%)	
Robson category 8	6 (1.3%)	
Robson category 9	1 (0.2%)	
Robson category 10	27 (5.9%)	

* Shapiro–Wilk test for normality distribution of continuous variables.

**Table 3 jcm-11-06503-t003:** Maternal characteristics and perinatal outcome of included COVID-19 pregnancies, according to the time period. Data presented as n (%) or median (interquartile range).

	Mar–Dec 2020 (n = 222)	Jan–April 2021 (n = 134)	May–Nov 2021 (n = 101)	*p* *
Maternal age (years)	30 [7]	30 [9]	31 [8]	0.890
Maternal pregestational BMI (m^2^/kg)	30.2 [5.1]	30.3 [7.6]	29.9 [7.1]	0.999
Maternal GWG (kg)	12 [5]	12 [7]	10 [6]	0.330
FGR	9 (4.1%)	10 (7.5%)	5 (5%)	0.373
Maternal hyperglycemia	16 (7.2%)	12 (9%)	8 (7.9%)	0.839
Maternal hypertension	12 (5.9%)	11 (8.2%)	7 (6.9)	0.577
pPROM	3 (1.4%)	4 (4%)	2 (2%)	0.560
Induction of labor	8 (3.6%)	12 (9%)	11 (10.9%)	0.027
GA at delivery (weeks)	39 [1.71]	38.6 [2.29]	38.9 [2.29]	0.021
Preterm birth	24 (10.9%)	20 (15.3%)	21 (20.8%)	0.017
Neonatal weight < 2500 g	28 (12.6%)	9/134 (6.7%)	11/101 (10.9%)	0.412
Neonatal weight < 1500 g	4 (1.8%)	3 (2.2%)	6 (5.9%)	0.057
Delivery by CS	152 (68.5%)	81 (60.4%)	58 (57.4%)	0.039
Urgent CS for respiratory distress	8 (3.6%)	4 (3%)	10 (9.9%)	0.032
CS Robson category 1	48/80 (60%)	27/47 (57.4%)	8/24 (33.3%)	0.043
CS Robson category 2	5/8 (62.5%)	4/7 (57.1%)	5/8 (62.5%)	0.971
CS Robson category 3	13/42 (31%)	6/33 (18.2%)	2/22 (9.1%)	0.037
CS Robson category 4	2/3 (66.7%)	0/2 (0%)	1/5 (20%)	0.230
CS Robson category 5	71/73 (97.3%)	34/34 (100%)	26/26 (100%)	0.244
CS Robson category 6	2/2 (100%)	2/2 (100%)	4/4 (100%)	N.A.
CS Robson category 7	1/1 (100%)	/	/	N.A.
CS Robson category 8	3/3 (100%)	0/1 (0%)	2/2 (100%)	0.853
CS Robson category 9	/	/	1/1 (100%)	N.A.
CS Robson category 10	7/10 (70%)	8/8 (100%)	9/9 (100%)	0.038

* Continuous variables were compared by Kruskal–Wallis test; categorical variables were compared by Mantel–Haenszel test for trend. / = absence of cases. N.A. = not applicable.

## Data Availability

The data that support the findings of this study are available from the corresponding author upon request.

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
