# Peer review of "Trends in Caesarean Section Rate According to Robson Group Classification among Pregnant Women with SARS-CoV-2 Infection: A Single-Center Large Cohort Study in Italy"

_jcm, 2022, doi:10.3390/jcm11216503_

Round 1

Reviewer 1 Report

The authors have analyzed how the indications for cesarean delivery changed throughout the COVID-19 pandemic. To the best of our knowledge, this is the first study to do so and the findings of this study provide an insight to the accumulating experience of the obstetricians dealing with pregnant women who have COVID-19. I thank the authors for their high-quality paper and recommend that their manuscript can be accepted for publication in Journal of Clinical Medicine after it has been edited for English. I would only recommend that the authors may mention the lack of a control group (delivering women without COVID-19) for comparing the alterations in indications for cesarean delivery as a limiting factor their findings.

Author Response

We thank the reviewer for the suggestion. We included the lack of a control group without COVID-19 in the Discussion section (Page 9 Line 271). We also checked for English language throughout the manuscript.

Reviewer 2 Report

The very topic seems to of the publication seems to be interesting. Currently, SARS-CoV-2 infection is one of the most challenging issues in modern medicine, especially in obstetrics and gynecology. The authors present a study that show the variability of cesarean section rate among pregnant women with SARS-CoV-2 infection during COVID-19 pandemics. It seems interesting that a fairly representative group of women was selected for the study. Additionally, it contains very adequate references supporting the presented theses. However, I have a few concerns about the paper. The diagrams presented in the paper are not readable - they should be redrafted in order to clearly present the results. Additionally, there are some grammar mistakes that should be corrected.

Author Response

We thank the reviewer for suggestions. We changed Figure 1 and Figure 2 to make them more readable. Furthermore, we checked English grammar throughout the manuscript

Reviewer 3 Report

1. Author claimed “Data from this study confirm that mode of delivery should not be influenced by the presence of COVID-19. With time, delivery management of infected pregnant women without respiratory distress results in CS rates similar to that reported for non-infected women.:  Those sentences give a little confusion to the reader since with high mobility and mortality during the initial covid-19 pandemic period, the delivery method could tend to C/S due to the fear of infection to health workers and the high mobility of pregnant women, in the late pandemic period with low mobility and mortality due to the virus nature, pregnant women should be managed as obstetric indications for C/S.  

2. In table 2, how the p values be calculated and what they mean?

3.Why the rate of urgent CS for respiratory distress (mother?) was significant higher in third period? And the induction rate different among the three period.

4. Every table and figure should be readable independently, so aberrations should be explained in every tables and figures.

Author Response

  1. Author claimed “Data from this study confirm that mode of delivery should not be influenced by the presence of COVID-19. With time, delivery management of infected pregnant women without respiratory distress results in CS rates similar to that reported for non-infected women.:  Those sentences give a little confusion to the reader since with high mobility and mortality during the initial covid-19 pandemic period, the delivery method could tend to C/S due to the fear of infection to health workers and the high mobility of pregnant women, in the late pandemic period with low mobility and mortality due to the virus nature, pregnant women should be managed as obstetric indications for C/S.  
  • We thank the reviewer for the considerations. Our cohort was enrolled from the beginning of the pandemic in Italy (March 2020) until November 2021, when the SARS-CoV-2 infection was more widespread but still associated with high morbidity and mortality due to the delta variant. In fact, presented data showed an increase in the urgent CS for maternal respiratory distress in the period 3 (Table 2, and Page 9 Line 247). Our hypothesis is that the decrease is CS section rate throughout the pandemic was associated with more confidence and less fear of healthcare personnel in the management of pregnancies complicated by COVID-19 (Page 9 Line 239), which can be a good parameter to highlight how healthcare for patients with COVID-19 improved over time.
  1. In table 2, how the p values be calculated and what they mean?
  • We thank the reviewer for this comment. We modified the Table legend including statistical tests used for the calculation. A p value <0.05 was considered as significant (Page 3 Line 141).

3.Why the rate of urgent CS for respiratory distress (mother?) was significant higher in third period? And the induction rate different among the three period.

            We thank the reviewer for the comment. Our data showed that induction of labor significantly increased over time from 3.6% to 10.9%, thus confirming the increased confidence of healthcare personnel in managing infected pregnant women. Data from our cohort also highlighted an increased CS rate in Robson group 10 in the period from May to November 2021, coincident with an increase in preterm birth and urgent CS performed for worsening maternal respiratory distress. These results are consistent with the increased morbidity observed in pregnancies with COVID-19 during the recent surge associated with the Delta variant, particularly in a pregnant population where vaccine acceptance was low at time. (Page 4 Line 151, Page 9 Lines 249-257).

  1. Every table and figure should be readable independently, so aberrations should be explained in every tables and figures.
  • We thank the reviewer for the precious suggestions. We modified Table and Figure legends consequently, in order to improve the presentation of data and analyses (See Figure 2, Table 3)